# Systematic Review and Meta-Analysis of Integrated Studies on Salmonella and Campylobacter Prevalence, Serovar, and Phenotyping and Genetic of Antimicrobial Resistance in the Middle East—A One Health Perspective

**DOI:** 10.3390/antibiotics11050536

**Published:** 2022-04-19

**Authors:** Said Abukhattab, Haneen Taweel, Arein Awad, Lisa Crump, Pascale Vonaesch, Jakob Zinsstag, Jan Hattendorf, Niveen M. E. Abu-Rmeileh

**Affiliations:** 1Swiss Tropical and Public Health Institute, Kreuzstr. 2, CH-4123 Allschwil, Switzerland; lisa.crump@swisstph.ch (L.C.); jakob.zinsstag@swisstph.ch (J.Z.); jan.hattendorf@swisstph.ch (J.H.); 2University of Basel, Petersplatz 1, CH-4001 Basel, Switzerland; 3Institute of Community and Public Health, Birzeit University, West Bank P.O. Box 14, Palestine; htaweel@birzeit.edu (H.T.); areenjaawwad@gmail.com (A.A.); nrmeileh@birzeit.edu (N.M.E.A.-R.); 4Department of Fundamental Microbiology, University of Lausanne, Bâtiment Biophore, CH-1015 Lausanne, Switzerland; pascale.vonaesch@unil.ch

**Keywords:** Middle East, One Heath, antimicrobial resistance, foodborne pathogens, *Campylobacter* spp., *Salmonella* spp., systematic review, meta-analysis

## Abstract

**Background:***Campylobacter* and *Salmonella* are the leading causes of foodborne diseases worldwide. Recently, antimicrobial resistance (AMR) has become one of the most critical challenges for public health and food safety. To investigate and detect infections commonly transmitted from animals, food, and the environment to humans, a surveillance–response system integrating human and animal health, the environment, and food production components (iSRS), called a One Health approach, would be optimal. **Objective**: We aimed to identify existing integrated One Health studies on foodborne illnesses in the Middle East and to determine the prevalence, serovars, and antimicrobial resistance phenotypes and genotypes of *Salmonella* and *Campylobacter* strains among humans and food-producing animals. **Methods**: The databases Web of Science, Scopus, and PubMed were searched for literature published from January 2010 until September 2021. Studies meeting inclusion criteria were included and assessed for risk of bias. To assess the temporal and spatial relationship between resistant strains from humans and animals, a statistical random-effects model meta-analysis was performed. **Results**: 41 out of 1610 studies that investigated *Campylobacter* and non-typhoid *Salmonella* (NTS) in the Middle East were included. The NTS prevalence rates among human and food-producing animals were 9% and 13%, respectively. The *Campylobacter* prevalence rates were 22% in humans and 30% in food-producing animals. The most-reported NTS serovars were *Salmonella* Enteritidis and *Salmonella* Typhimurium, while *Campylobacter jejuni* and *Campylobacter coli* were the most prevalent species of *Campylobacter*. NTS isolates were highly resistant to erythromycin, amoxicillin, tetracycline, and ampicillin. *C. jejuni* isolates showed high resistance against amoxicillin, trimethoprim–sulfamethoxazole, nalidixic acid, azithromycin, chloramphenicol, ampicillin, tetracycline, and ciprofloxacin. The most prevalent Antimicrobial Resistance Genes (ARGs) in isolates from humans included tetO (85%), Class 1 Integrons (81%), blaOXA-61 (53%), and cmeB (51%), whereas in food-producing animals, the genes were tetO (77%), Class 1 integrons (69%), blaOXA-61 (35%), and cmeB (35%). The One Health approach was not rigorously applied in the Middle East countries. Furthermore, there was an uneven distribution in the reported data between the countries. **Conclusion**: More studies using a simultaneous approach targeting human, animal health, the environment, and food production components along with a solid epidemiological study design are needed to better understand the drivers for the emergence and spread of foodborne pathogens and AMR in the Middle East.

## 1. Introduction

*Campylobacter* spp. and *Salmonella* spp. are the leading causes of foodborne diseases worldwide [1,2]. According to a report published by the World Health Organization (WHO) in 2018, the global burden of food-borne illnesses is 1 in 10 individuals each year [3]. Annually, non-typhoid *Salmonella* (NTS) is responsible for more than 155,000 annual deaths and 94 million annual cases worldwide [4]. *Campylobacter* infection is a public health problem, causing about 8% of global diarrheal cases [5]. Since 2005, *Campylobacter* has been the most reported gastrointestinal bacterial pathogen in humans in the European Union (EU) [6,7].

The Middle East region has the third-highest prevalence of foodborne illness, with 100 million people estimated to be ill from foodborne illnesses each year. Norovirus, *Escherichia coli*, *Campylobacter*, and NTS are responsible for 70% of all foodborne diseases in the Middle East region [8]. The incidence rate of NTS among Jordanians was 124 per 100,000 in 2003–2004 and 30 per 100,000 among Israelis in 2009 [9,10]. In addition, *Campylobacter* was identified in 61% of children with dysentery (63/99) in Israel, 33% (76/230) in Iran 4.7% (7/150) in Palestine, and 3.7% (13/356) in Egypt during the period 2005–2015 [11,12,13,14].

Antimicrobial resistance (AMR) is a major public health concern mainly resulting from the use and misuse of antimicrobial agents. AMR occurs when bacteria, fungi, parasites, and viruses change over time and are no longer susceptible to medicines, making infections difficult to treat and increasing the risk of spreading the infection, intensifying the severity of the disease, and raising death rates [15,16]. After the bacteria has acquired resistance, AMR disseminates by clonal spreads of the bacteria and horizontal gene transfer (HGT), that is, by integrons or plasmids, leading to the accumulation of antimicrobial resistance genes (ARGs) in pathogenic and non-pathogenic bacteria within an individual organism [16]. Rising antimicrobial use contributes to the sharing of resistant bacteria and resistance genes between food animals and humans through the food production chain [15]. AMR in *Campylobacter* spp. and *Salmonella* spp. has been shown to be directly associated with antimicrobial use in animal production. Food-borne diseases caused by these resistant bacteria are well documented in humans [15].

Since humans and animals are in close contact and are intricately interconnected, food safety and AMR are fundamental One Health issues [17,18]. However, most of the current research in low- and middle-income countries (LMICs) focuses on human or animal health risks separately and only a few studies have been conducted to understand the problem in an interconnected manner [19,20]. Additional components of human and animal health must be incorporated to make significant progress in reducing many foodborne diseases [19].

The Joint Programming Initiative on Antimicrobial Resistance (JPIAMR) (www.jpiamr.eu, accessed on 19 March 2022) identified several critical knowledge gaps. First, the relative contributions of different sources of antibiotics and antibiotic-resistant bacteria into the environment are unmeasured. Second, the role of the environment, particularly the anthropogenic inputs, on the evolution of resistance is not understood. Third, the overall human and animal health impacts caused by exposure to resistant bacteria from the environment have not been studied. Finally, the efficacy of technological, social, economic, and behavioral interventions to mitigate environmental antibiotic resistance have not been evaluated [21]. A recent review of integrated studies on antimicrobial resistance in Africa concluded that data on AMR from a One Health perspective in Africa are scarce with only 18 studies meeting the minimal standards of addressing simultaneously at least two of the environment–animal–human realms [16].

This systematic review and meta-analysis aims to summarize the scientific literature published between January 2010 and August 2021 on the prevalence, serovars, and antimicrobial resistance phenotypes and genotypes (ARGs) of *Salmonella* and *Campylobacter* strains from integrated studies, studying at the same time humans and food-producing animals and their products in the Middle East region. In addition, it attempts to address the knowledge gap and summarize the available information about the situation by applying the integrated studies to follow up *Salmonella* spp. and *Campylobacter* spp. as the leading foodborne illnesses in the Middle East.

## 2. Methodology

The protocol for this systematic review was registered in the International Prospective Register of Systematic Reviews (PROSPERO ID: CRD42021277400).

### 2.1. Search Strategy

We conducted a systematic search on PubMed, Web of Science, and Scopus, limiting the search to the literature published from 2010 until 30 September 2021. Two reviewers performed the initial search, abstract screening, and data extraction, and any discordances were solved by a third reviewer. The exact search strategy used for each database is included in Appendix A.

### 2.2. Inclusion and Exclusion Criteria

We aimed to analyze the available information about prevalence, serovar distribution, and antimicrobial resistance phenotypes and genotypes of *Salmonella* and *Campylobacter* strains among humans and food-producing (terrestrial) animals and their products in the Middle East region. It included all peer-reviewed literature published from 1 January 2010, until 30 September 2021. The search included only studies that were published in English. We excluded publications published before 2010, grey literature, non-peer-reviewed literature, and studies with a different design than cross-sectional, cohort studies, and studies using survey system data (Routine data). In addition to information on Salmonella spp. and Campylobacter spp. isolates originating from companion animals, plant-based food, aquatic products (fish), water sources, and concerning *Salmonella* enterica serotypes Typhi and Paratyphi.

### 2.3. Study Selection

Two independent reviewers used Covidence software (www.covidence.org, accessed on 23 September 2021) for the title and abstract screening. Studies that were eligible for full-text review were further reviewed. Subsequently, risk assessment and data extraction were undertaken. Disagreements between reviewers in the title and abstract screening or full-text review were resolved through consultation with a third reviewer.

### 2.4. Data Extraction

Two independent reviewers extracted the data for the included papers, and the required data was entered into an Excel (Microsoft Inc.^TM^, Redmond, WA, USA) sheet. Data included author, publication year, year of data collection, collection country, study outcomes, study design, the validity and reliability of the study methodology, as well as details available regarding analysis, human and animal sample sizes, sample sources, isolated bacteria source, and prevalence. In addition, data regarding serotype prevalence, AMR gene prevalence, and NTS and *Campylobacter* AMR profiles were collected.

### 2.5. Risk of Bias Assessment

We used the risk of bias tool developed by Hoy et al., 2012 [22] to assess the overall quality of the papers. Two independent reviewers performed the risk of bias assessment and disagreements were solved by consensus.

### 2.6. Data Synthesis

The number of studies remaining at each stage of the selection process is summarized in the flowchart in Figure 1.

The pooled prevalence rate of *Salmonella* spp. and *Campylobacter* spp. and their main serotypes for human and food-producing animals (live animals and products) were calculated separately based on the following Equation (1):(1)$$Prevalence\;rate=\frac{\;No.\;of\;isolated\;bacteria}{Total\;number\;of\;collected\;samples}.$$

AMR profile among NTS and *C. jejuni* was calculated using Equation (2):(2)$$Resistance\;rate=\frac{No.\;of\;resistance\;Isolates}{Total\;number\;of\;isoalted\;bacteria}.$$

### 2.7. Statistical Analysis

Relative risks were assessed based on the total number of samples and the number of NTS, *Campylobacter* spp., and AMR positive samples (phenotype and genotype). Studies were stratified by bacterial species and sources. A pooled risk ratio (RR) was calculated separately for each bacterial species. The *I*^2^ and *r*^2^ statistics assessed heterogeneity. We exclusively used the random-effects model, irrespective of the heterogeneity results. For all statistical analyses, we used the R software environment version 4.0.3 and the “meta-package” version 4.14-0. We used the function ‘metabin’ using the Mantel–Haenszel method with inverse variance weighting for pooling [23].

## 3. Results

### 3.1. Studies Identified and Included in the Final Analysis

Based on the eligibility criteria, a total of 2534 publications were identified. After removing duplicates, we screened 1610 abstracts of which 565 were eligible for full-text screening. Out of 565 articles, 41 studies met the inclusion criteria for this meta-analysis (Figure 1). In total, 31 studies used a cross-sectional study design, and 10 studies used routine data (Appendix A).

The overall result of the risk assessment that was conducted for the included studies indicated that the majority of studies had an overall low risk of bias, and none of the papers had a high risk of bias. This result was based on the risk of bias assessment using the Hoy et al., 2012 tool [22].

### 3.2. Overview of the Selected Studies

A total of 16 countries were included in this literature review: Qatar, United Arab Emirates, Bahrain, Saudi Arabia, Kuwait, Israel, Oman, Iran, Jordan, Lebanon, Palestine, Syria, Yemen, Turkey, Iraq, and Egypt. Of these, nine countries had no published literature matching the study inclusion criteria available (Qatar, United Arab Emirates, Bahrain, Saudi Arabia, Kuwait, Oman, Syria, Yemen, and Iraq), while seven countries had at least one article available (Egypt, Iran, Turkey, Lebanon, Israel, Jordan, and Palestine).

Of the included studies, 26 (63.41%) were conducted in Egypt, 8 in Iran (19.51%), 2 in Turkey (4.88%), 2 in Lebanon (4.88%), 1 in Israel (2.44%), 1 in Jordan (2.44%), and 1 in Palestine (2.44%) (Figure 2a). Of these, 17 reports (42%) included data about *Salmonella* spp. (the number of reports used for *Salmonella* spp is the same as that used for NTS), and 8 reports (20%) had data about *Campylobacter* spp. In addition, some articles focused on one of *Salmonella* and *Campylobacter* serovars; *Campylobacter jejuni* (nine reports, 22%), *Campylobacter coli* (one report, 2%), *Salmonella* Enteritidis (four reports, 10%), *Salmonella* Typhimurium (one report, 2%), and *Salmonella* Heidelberg (one report, 2%) (Figure 2b) and (Appendix A).

### 3.3. Prevalence and Serotype Distribution of Salmonella spp. and Campylobacter spp. among Humans and Food-Producing Animals

Out of 41 eligible articles, 31 were cross-sectional studies. We used the cross-sectional data to calculate the prevalence rate for each pathogen separately. Of the 1317 human samples, 167 (13%) were positive for *Salmonella* spp. (14% in diarrhea patients and 9% in high-risk population). In food-producing animals, out of 3520 samples, 585 (17%) were positive for *Salmonella* spp. (31% in poultry and poultry products and 4% in ruminants and ruminant products). Moreover, NTS was reported with a prevalence of 9% (109/1167) in humans (10% in diarrhea patients and 6% in high-risk populations) and 13% (352/2718) in food-producing animals (33% in poultry and poultry products and 4% in ruminants and ruminant products). The two most common NTS serovars were *S.* Typhimurium with a prevalence of 5% (36/780) in humans (4% in diarrhea patients and 6% in high-risk populations) and 3% (91/3038) in food-producing animals (7% in poultry and poultry products and 0.6% in ruminants and ruminant products) and *S.* Enteritidis with a prevalence of 2% (12/585) in humans (2% in diarrhea patients and 2% in high-risk population) and 3% (87/2534) in food-producing animals (9% in poultry and poultry products and 0.3% in ruminants and ruminant products) (Table 1 and Table 2).

*Campylobacter* spp. was reported with a prevalence of 22% (435/2008) in humans (23% in diarrhea patients and 14% in high-risk populations) and 30% (1253/4122) in food-producing animals (39% in poultry and poultry products and 10% in ruminants and ruminant products). The two most commonly detected *Campylobacter* spp. serovars were *C. jejuni* with a prevalence of 16% (422/2693) in humans (16% in diarrhea patients and 9% in high-risk populations) and 22% (1182/5472) in food-producing animals (25% in poultry and poultry products and 14% in ruminants and ruminant products) and *Campylobacter coli* with a prevalence of 4% (72/1938) in humans (3% in diarrhea patients and 8% in high-risk populations) and 9% (367/4037) in food-producing animals (13% in poultry and poultry products and 2% in ruminants and ruminant products) (Table 1 and Table 2).

### 3.4. Microbial Resistance Patterns Detected by Phenotypic Screening

Based on the prevalence rate results and the number of eligible articles included in this review, NTS and *C. jejuni* were the two most prevalent representatives of *Salmonella* spp. and *Campylobacter* spp., respectively. The average resistance of NTS and *C. jejuni* was calculated for each pathogen separately depending on the source of the isolated bacteria (human or food-producing animals) (Appendix A).

For NTS, information on 13 different antibiotics was available and is summarized in Table 3. NTS isolated from humans showed resistance against erythromycin (100%), amoxicillin (71%), tetracycline (62%), ampicillin (52%), azithromycin (43%), amoxicillin–clavulanic acid (42%), streptomycin (40%), cefotaxime (31%), trimethoprim–sulfamethoxazole (24%), chloramphenicol (15%), ciprofloxacin (9%), imipenem (2%), and ceftriaxone (1%). NTS isolated from food-producing animals showed resistance against erythromycin (100%), amoxicillin (91%), tetracycline (50%), ampicillin (69%), azithromycin (9%), amoxicillin–clavulanic acid (70%), streptomycin (43%), cefotaxime (63%), trimethoprim–sulfamethoxazole (8%), chloramphenicol (12%), ciprofloxacin (17%), and ceftriaxone (7%). Amoxicillin–clavulanic acid was used more frequently in animal isolates (70%), with a pooled risk ratio (RR) of 1.09 (95% confidence interval (CI): 1.01–1.18) and with a heterogeneity of *I*^2^ = 34% and *r*^2^ ≤ 0.001, while the pooled RR close to 1 in ampicillin and streptomycin suggests a similar probability of occurrence in humans and animals. For the other antibiotics, no clear pattern was detected (Table 3).

For *C. jejuni,* we had data on 11 antibiotics. The phenotypic resistance results are summarized in (Table 4). *C. jejuni* isolated from humans showed resistance against amoxicillin (100%), trimethoprim–sulfamethoxazole (93%), nalidixic acid (89%), azithromycin (88%), chloramphenicol (82%), ampicillin (81%), tetracycline (75%), ciprofloxacin (73%), amoxicillin–clavulanic acid (68%), erythromycin (65%), and streptomycin (39%). *C. jejuni* isolated from food-producing animals showed complete resistance against amoxicillin (100%) and azithromycin (100%) and to a lesser extent resistance against trimethoprim–sulfamethoxazole (83%), nalidixic acid (76%), chloramphenicol (69%), ampicillin (64%), tetracycline (56%), ciprofloxacin (71%), amoxicillin–clavulanic acid (32%), erythromycin (38%), and streptomycin (21%).

Azithromycin was detected more frequently in animal isolates, with a pooled risk ratio (RR) of 1.13 (95% confidence interval (CI): 1.04–1.24) and a heterogeneity of *I*^2^ = 72% and *r*^2^ = 0.033. Amoxicillin–clavulanic acid was detected more frequently in human isolates, with a pooled risk ratio (RR) of 0.79 (95% confidence interval (CI): 0.67–0.95) and heterogeneity of *I*^2^ = 53% and *r*^2^ ≤ 0.001. For the other antibiotics, no clear pattern was detected. Most phenotypic resistance had a pooled RR close to 1, suggesting a similar probability of occurrence in humans and animals (Table 4)

### 3.5. Assessment of Shared Antimicrobial Resistance Genes

Only six studies reported resistance genes targeted at three serovars of *Salmonella* spp. and *Campylobacter* spp. These serovars were *C. jejuni* (three studies), NTS (two studies), and *Salmonella Heidelberg* (one study). We calculated the average prevalence for every single resistance gene from food-producing animals and human sources separately. For human isolates, *tetO* was the gene with the highest prevalence (85%), followed by Class 1 Integrons (81%), *blaOXA-61 (53%)*, *cmeB (51%)*, *blaCMY-2 (38%),* Class 2 integrons (29%), *tetA (21%), blaOXA (21%), blaSHV (19%), AAC(6’)-Ib (16%), blaCTXM-1 (16%), blaAMPc (13%),* and *blaTEM (13%)*. For food-producing animals, tetO was the most prevalent gene (77%), followed by Class 1 Integrons (69%), *blaOXA-61 (35%), cmeB (35%), tetA (30%),* Class 2 integrons (27%)*, blaCTXM-1 (22%), AAC(6’)-Ib (22%), blaSHV (20%), blaTEM (15%), blaCMY-2 (11%), blaOXA (11%),* and *blaAMPc (3%)* (Table 5) (Appendix A).

Resistance in *Campylobacter* spp. was exclusively reported as data for *C. jejuni* isolates. The three studies reporting data on resistance compromised 274 isolates (232 human isolates and 42 food-producing animals and their products). The most frequent genes were Class 1 Integrons (96%), *tetO* (85%), *blaOXA-61* (53%), *cmeB* (51%), and *tetA* (17%). For food-producing animals and their product isolates, the most frequently detected genes were Class 1 Integrons (100%), *tetO* (77%), *blaOXA-61* (35%), *cmeB* (35%), and *tetA* (30%). There was no evidence for a significant difference in the occurrence of the genes between human and food-producing animals and their products (Table 5) except for Class 1 integrons, which were detected more frequently in food-producing animals, with a risk ratio (RR) of 1.04 (95% confidence interval (CI): 1.01; 1.08) (Table 5). No clear pattern was detected for the other genes, with most of the genes having a pooled RR close to 1, suggesting a similar probability of occurrence in humans and animals (Table 5) (Appendix A).

The two studies on NTS compromised 197 isolates (125 human isolate and 72 food-producing animals and their products). The most frequent genes were Class 1 Integrons (51%), Class 2 Integrons (29%), *blaSHV* (16%), *blaCTXM-1* (16%), *AAC(6’)-Ib* (16%), *blaTEM* (10%), and blaAMPc (1%). For food-producing animal isolates, the most frequently detected genes were Class 1 Integrons (41%), Class 2 Integrons (27%), *blaSHV* (22%), *blaCTXM-1* (22%), *AAC(6’)-Ib* (22%), and *blaTEM* (15%). No clear pattern emerged for the majority of the genes in the random effect models comparing frequencies in humans and animals (Table 5), suggesting there was no evidence for a significant difference in the occurrence of the genes between humans and food-producing animals (Table 5) (Appendix A).

The single study including *Salmonella* Enterica Serovar Heidelberg compromised 33 isolates (24 human isolates and 9 food-producing animals and their products). In isolates from human sources, the most frequent genes were *blaAMPc* (50%), *blaCMY-2* (38%), *blaTEM* (29%), blaSHV (25%), and blaOXA (20%). For food-producing animal isolates, the most frequently detected genes were *blaAMPc* (11%), *blaCMY-2* (11%), *blaTEM* (11%), *blaSHV* (11%), and *blaOXA* (11%). There was no evidence of a significant difference in the occurrence of the genes between humans and food-producing animals (Appendix A).

## 4. Discussion

Although 41 articles were eligible for inclusion in this systematic review and meta-analysis, there is an uneven distribution of the sources of the studies included. The majority (63%) of eligible studies were from Egypt and 20% from Iran. On the other hand, there are no published papers on applying a comprehensive One Health approach to study one of the two major foodborne diseases (*Salmonella* and *Campylobacter*) in 9 counties from the 16 Middle Eastern countries, and none came from those high-income countries members of the Gulf Cooperation Council.

Of the 41 studies included in this review, 31 were cross-sectional, and 10 were routine data studies. Studies allow a comparison between human and animal sources; they do not evaluate actual transmission methods because the few studies eligible for inclusion in this review suffered from insufficient statistical data on foodborne pathogens and AMR and assess only selected sections of the social ecosystem.

Furthermore, our systematic review and meta-analysis showed the prevalence of *Salmonella* spp. and *Campylobacter* spp., resistance rates, and antimicrobial resistance genes circulating in the Middle East region by using the random-effects model. The model showed high heterogeneity results, which indicate variability in the study data. This might be due to the study design (epidemiological study vs. routine data) or due to diverse sample types. The human isolates used in the studies were from different sources (symptomatic and asymptomatic participants), and the animal isolates used were from various sources (live animals and products). Finally, the small sample size in each study and, in particular, the human sample size could influence the results when measuring the prevalence and the relationship between the humans and animal settings. The heterogeneity might explain the insignificant relationship between animals and humans.

The low quantity (low sample size), uneven distribution in the reported data, and weak epidemiological study designs from a One Health methodological perspective [24] in the studies that targeted foodborne illness and antimicrobial resistance in the Middle East can be explained by the food safety system’s challenges in this region. These challenges are the lack of epidemiological and disease ecological capacity, diagnostic tools, and laboratory facilities. Moreover, there is a lack of quality control and standardization of microbiological identification and susceptibility testing techniques [15].

Our review demonstrates the prevalence of NTS and *Campylobacter* spp. and their serovars circulating in the Middle East. The pooled prevalence of *Campylobacter* spp. among humans was close to the higher estimate for the ranges reported in Sub-Saharan Africa: and Northern Africa (2–27.5%) and more than the ranges reported in Southeast Asia (8%) [25,26,27]. Additionally, the results show the prevalence of *Campylobacter* spp. in food-producing animals and their products (30%). For *Campylobacter* spp., the prevalence rate is similar to the systematic review and meta-analysis results that targeted *Campylobacter* spp. globally, with approximately 30% of animal food products analyzed reporting *Campylobacter* spp. [28]. Additionally, we looked at which *Campylobacter* serovars are circulating in the Middle East and found *C. jejuni* and *C. coli* to be the predominant serovars, similar to results that targeted *Campylobacter* in Africa as the *C. jejuni* and *C. coli* predominates in Sub-Saharan Africa [29].

We identified two systematic reviews conducted by Al-Rifai and his colleagues that targeted the Middle East and South African countries in 2019 and 2020; we will compare our results with these relevant studies. In this review, the pooled prevalence rates of NTS were 9% and 13% among humans and animals and their products, respectively. The pooled prevalence in humans is higher than the results in the Al-Rifai study (2019) which was 7% [30]. In addition, the prevalence of the food-producing animals in this review is more than the results of the Al-Rifai (2020) study, which was 9% [31]. Our findings are similar to Al-Rifai’s studies of NTS serovars, in which *S.* Typhimurium and *S.* Enteritidis were the main NTS serovars reported in this region [30,31].

Furthermore, this systematic review showed *Campylobacter* and *Salmonella* serovars are highly prevalent in poultry and poultry products in the Middle East. The Campylobacter prevalence in animals was less than the prevalence reported in broiler meat in Poland, Slovenia, Spain, and Austria. Conversely, more than reported in Denmark and Finland [32]. The *Salmonella* prevalence in animals showed results less than the prevalence reported in raw chicken at retail markets in China and more than reported in chicken carcasses in Spain [33,34]. This endemic *Campylobacter* and *Salmonella* bacteria in animal food products can be explained, at least partially, by the changes in animal production systems that have tended to be more intense over the past decades [28]. These findings are essential because transmission along the production chain is generally established as the most common pathway used by *Campylobacter* and *Salmonella* to generate human infection [29].

AMR is a transboundary public health problem. New types of AMR strains can expand worldwide following initial endemic emergence, as demonstrated by several resistant pathogens that spread globally [35]. Our meta-analysis revealed a high NTS resistance against erythromycin, amoxicillin, tetracycline, and ampicillin for isolates from humans and food-producing animals. The isolates have similar resistance rates between humans and animals in erythromycin but are higher in isolates from animal sources for amoxicillin and ampicillin and higher in isolates from human sources in tetracycline. These results are close to those reported by Alsayeqh’s systematic review in the Middle East region [15]. In addition, the most recent report on AMR in the EU in 2019–2020 found that resistance of NTS to sulfonamides, ampicillin, and tetracycline was high in human isolates, while it ranged from moderate to very high in animal isolates [36].

AMR phenotypic results for *C. jejuni* isolates (human and food-producing animals) showed high resistance against amoxicillin, trimethoprim–sulfamethoxazole, nalidixic acid, azithromycin, chloramphenicol, ampicillin, tetracycline, and ciprofloxacin. These findings were close to Alsayeqh’s systematic review for trimethoprim–sulfamethoxazole, nalidixic acid, and tetracycline. In comparison, it has a lower resistance rate for amoxicillin, chloramphenicol, ampicillin, and ciprofloxacin [15]. Our results show that *C. jejuni* isolated from humans has a phenotypical resistance rate against nalidixic acid and tetracycline more than that reported in Italy and less against ciprofloxacin based on the same study results [37]. At the same time, our results show that *C. jejuni* isolated from animals has a phenotypical resistance rate against nalidixic acid, ciprofloxacin, and tetracycline more than that reported in broiler chicken in Belgium [38]. This systematic review demonstrated moderate to high resistance of *C. jejuni* to erythromycin. Conversely, the recent EU report on AMR found that *C. jejuni* resistance to erythromycin was either undetected or detected at very low levels in *C. jejuni* from food-producing animals and humans [36].

The WHO, Food and Agriculture Organization of the United Nations (FAO), and World Organization for Animal Health (OIE) recommend reducing antibiotic use in animal husbandry, particularly for those known to cause cross-resistance [39,40,41]. However, some antimicrobials traditionally used in animal production as growth promoters and/or for treating gastrointestinal infections are also used to control human infectious diseases (e.g., tetracycline and quinolones) [42]. The misuse and overuse of antimicrobials in clinical and veterinary medicine and agriculture have increased antimicrobial resistance pathogens, including *Campylobacter* and *Salmonella* [43]. For instance, the Quesada study showed that *Salmonella* isolated from animal food has significant antibiotic resistance in Latin American countries [44]. The therapeutic and prophylactic use of antibiotics in animal production for long periods is likely contributing to the widespread resistance against antibiotics [43]. More integrated environmental–animal–human studies are needed in the region to ascertain its effect on public health. This way, microbiological and clinical evidence on the transmission of AMR between animals and humans can be ascertained in Middle Eastern countries [43,44,45].

Data on antimicrobial resistance genes (ARGs) among *Salmonella* spp. and *Campylobacter* spp. in the Middle East is limited. However, based on the reported information, we can argue that food-producing animals and their products in the Middle East are not the main drivers for the emergence of ARGs.

Based on our eligibility criteria, six studies targeted ARGs among *Salmonella* and *Campylobacter* as foodborne illnesses in the Middle East region [46,47,48,49,50,51]. Besharati’s study and Youssef’s study were two studies that reported the ARGs among NTS. Besharati’s study shows an association between the AMR phenotype results and ARGs results in Integron 1 and 2 classes and trimethoprim/sulfamethoxazole in Iran. Conversely, in Youssef’s study, results from Egypt revealed no association between AMR phenotype results and ARGS.

Three studies reported the ARGs among *C. jejuni* (Abd-El-Aziz, Divsalar, and Ghoneim) [46,47,48]. The results in Divsalar and Ghoneim could not show a significant association between the targeted ARGs and the AMR phenotype results. In turn, Abd-El-Aziz found an association between Class 1 integrons and aminoglycoside resistance.

We identified small-scale studies with a small sample size for the ARGs in the NTS and *C. jejuni*. The small sample size in the eligible studies might be responsible for the insignificant difference in the occurrence of the genes between humans and food-producing animals. Our results agreed with Escher’s systematic review that targeted ARGs in Africa and found eligible studies characterized by small-scale studies and with a small sample size [16]. Therefore, future studies should have an integrated approach to assess the ARGs and should have a suitable sample size.

Partial sequencing of *C. jejuni* and NTS were performed using conventional PCR to extract the ARGs. Therefore, there is a lack of laboratory techniques that determine the order of bases in an organism’s genome in one process such as with Whole-genome sequencing (WGS), to follow the foodborne illnesses and ARGs. Undertaking WGS of isolates, especially those with high-level antibiotic resistance, is strongly encouraged to demonstrate the involved ARGs and their genetic localization (plasmid, chromosome, genomic islands, integrative and conjugative element, and transposon) as well as to detect the most prevalent resistant serovars [36,52,53], detail their potential of horizontal transmission, and evaluate the different sources and comparison of human and animal isolates [54].

## 5. Conclusions

To the best of our knowledge, this is the first systematic review assessing integrated environment–animal–human studies using a One Health approach in the Middle East to pursue foodborne illnesses and antimicrobial resistance. The One Health approach was not rigorously applied in the Middle East countries. In addition to weak epidemiological study designs from a One Health methodological perspective, there is an uneven distribution in the reported data with about 60% of Middle Eastern countries having no published papers included in this review. More research on foodborne illnesses and AMR in the Middle East is urgently needed. The AMR phenotype results showed a high prevalence of resistance rate for the isolated bacteria that highlights the importance of antimicrobial stewardship in humans and animals in tandem. Furthermore, introducing new laboratory techniques that determine the order of bases in an organism’s genome is essential to follow up the foodborne illness outbreak and ARGs.

A simultaneous approach that targets human and animal health in tandem with a solid epidemiological study design has a high potential to provide evidence for understanding the drivers for the emergence and spread of foodborne pathogens and AMR. A comprehensive One Health approach, integrating by a sound epidemiological design the spatio-temporal relationship of humans, animals, and their environment, will allow us to identify key transmission pathways, which are essential for designing more efficient food safety systems and AMR control policies. 

## Figures and Tables

**Figure 1 antibiotics-11-00536-f001:**
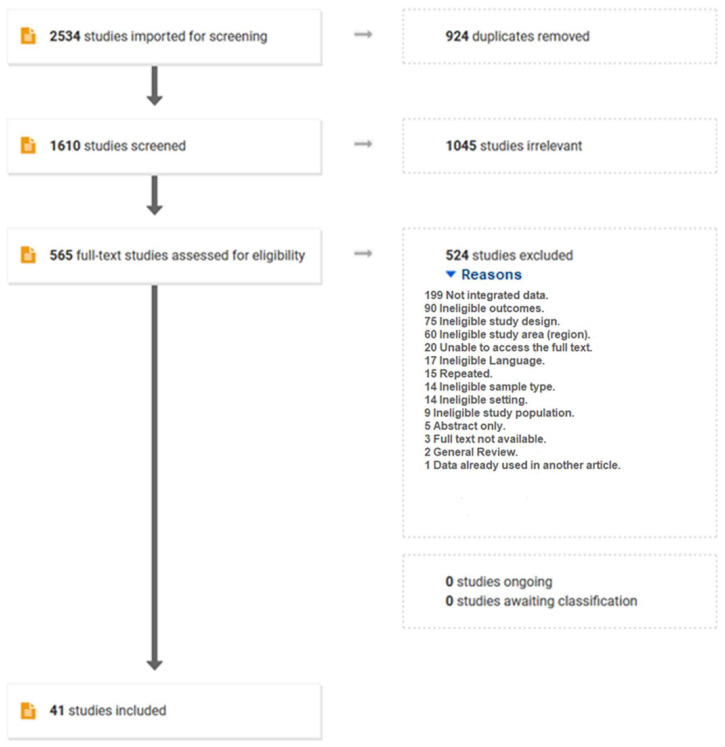
Search strategy and PRISMA flow diagram.

**Figure 2 antibiotics-11-00536-f002:**
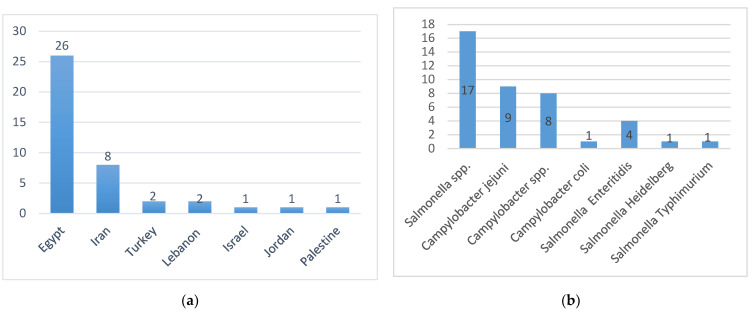
Number of studies (**a**) per country and (**b**) per pathogen.

**Table 1 antibiotics-11-00536-t001:** Overall prevalence of *Salmonella* and *Campylobacter* and main serotypes.

Pathogens	No. of Isolated Bacteria from Humans	Total Number of Collected Samples from Humans	The Pooled Prevalence Rate among Humans (%)	No. of Isolated Bacteria from Animals	Total Number of Collected Samples from Animals	The Pooled Prevalence Rate among Animals (%)
*Salmonella* spp.	167	1317	13	585	3520	17
nontyphoidal *Salmonella*	109	1167	9	352	2718	13
*S. typhimurium*	36	780	5	91	3038	3
*S. enteritidis*	12	585	2	87	2534	3
*Campylobacter*	435	2008	22	1253	4122	30
*C. jejuni*	422	2693	16	1182	5472	22
*C. coli*	72	1938	4	367	4037	9

**Table 2 antibiotics-11-00536-t002:** Prevalence of *Salmonella* and *Campylobacter* and main serotypes based on the samples sources.

Pathogens	N (%) Isolated Bacteria from Asymptomatic Humans	Total Number of Asymptomatic Humans Samples	N (%) Isolated Bacteria from Symptomatic Humans	Total Number of Symptomatic from Humans Samples	N (%) Isolated Bacteria from Poultry and Poultry Products	Total Number of Poultry and Poultry Products Samples	N (%) Isolated Bacteria from Ruminants and Ruminant Products	Total Number of Ruminants and Ruminant Products Samples
*Salmonella* spp.	29 (9%)	342	138 (14%)	975	492 (31%)	1597	76 (4%)	1717
Nontyphoidal *Salmonella*	11 (6%)	192	98 (10%)	975	259 (33%)	795	76 (4%)	1717
*S. typhimurium*	13 (6%)	205	23 (4%)	575	80 (7%)	1195	9 (0.6)	1637
*S. enteritidis*	1 (2%)	60	11 (2%)	525	82 (9%)	897	5 (0.3)	1637
*Campylobacter*	28 (14%)	206	407 (23%)	1802	1048 (39%)	2695	205 (10%)	1427
*C. jejuni*	21 (9%)	226	401 (16%)	2467	968 (25%)	3894	214 (14%)	1578
*C. coli*	18 (8%)	236	54 (3%)	1702	341 (13%)	2610	26 (2%)	1427

**Table 3 antibiotics-11-00536-t003:** Microbial resistance patterns detected by phenotypic screening among non-typhoidal *Salmonella*.

	Non-Typhoidal *Salmonella*							
Antibiotic	No. of Resistance Human Isolates	Human Isolates	Resistance Ratio/Human Isolates	No. of Resistance Animal Isolates	Animal Isolates	Resistance Ratio/Animal Isolates	Relative Risk	95%CI
Amoxicillin–Clavulanic acid	53	126	42%	62	88	70%	1.09	[1.01; 1.18]
Amoxicillin	50	70	71%	64	70	91%	4.02	[0.16; 103.61]
Ampicillin	97	186	52%	96	139	69%	1.10	[0.92; 1.31]
Azithromycin	32	75	43%	2	22	9%	0.21	[0.06; 0.82]
Cefotaxime	45	145	31%	58	92	63%	3	[0.23; 39.38]
Ceftriaxone	2	231	1%	9	131	7%	4.33	[0.93; 20.26]
Chloramphenicol	42	281	15%	21	181	12%	1.29	[0.86;1.96]
Ciprofloxacin	26	281	9%	30	181	17%	1.36	[0.73; 2.51]
Erythromycin	57	57	100%	52	52	100%	1	[0.96; 1.04]
Imipenem	3	194	2%	0	108	0%	0.45	[0.05; 4.02]
Streptomycin	50	126	40%	38	88	43%	1.09	[0.80; 1.49]
Tetracycline	142	231	62%	66	131	50%	0.79	[0.59; 1.06]
Trimethoprim–sulfamethoxazole	67	281	24%	15	181	8%	0.57	[0.18; 1.80]

**Table 4 antibiotics-11-00536-t004:** Microbial resistance patterns detected by phenotypic screening among *Campylobacter jejuni*.

	*Campylobacter jejuni*							
Antibiotic	No. of Resistance Human Isolates	Human Isolates	Resistance Ratio/Human Isolates	No. of Resistance Animal Isolates	Animal Isolates	Resistance Ratio/Animal Isolates	Relative Risk	95%CI
Amoxicillin–Clavulanic acid	283	416	68%	56	173	32%	0.79	[0.67; 0.95]
Amoxicillin	297	297	100%	52	52	100%	1	[0.96; 1.04]
Ampicillin	466	579	81%	142	223	64%	1	[0.97; 1.03]
Azithromycin	261	297	88%	52	52	100%	1.13	[1.04; 1.24]
Chloramphenicol	258	316	82%	50	73	69%	1.01	[0.89; 1.14]
Ciprofloxacin	460	627	73%	187	265	71%	0.92	[0.84; 1.01]
Erythromycin	393	608	65%	92	244	38%	1	[0.97; 1.03]
Nalidixic acid	558	627	89%	201	265	76%	0.89	[0.77; 1.02]
Streptomycin	213	544	39%	50	235	21%	1.02	[0.83; 1.26]
Tetracycline	248	330	75%	119	213	56%	0.94	[0.84; 1.05]
Trimethoprim–sulfamethoxazole	371	399	93%	85	103	83%	1.01	[0.97; 1.04]

**Table 5 antibiotics-11-00536-t005:** Prevalence of AMR genes found in non-typhoidal *Salmonella* spp. and *Campylobacter jejuni*.

AMR Gen	Study ID	Pathogen	HN	HI	Prevalance_H	AN	AI	Prevalnce_A_	RR	95%CI	Lab Technique
*blaAMPc*	Besharati et al., 2020 and Elhariri et al., 2020	NTS and S. H	99	13	13.13%	31	1	3.23%	0.34	[0.07; 1.72]	PCR
*AAC(6’)-Ib*	Youssef et al., 2021	NTS	50	8	16.00%	50	11	22.00%	1.38	[0.6; 3.13]	PCR
*bla CMY-2*	Elhariri et al., 2020	S. H	24	9	37.50%	9	1	11.11%	0.3	[0.04; 2.02]	PCR
*bla CTXM-1*	Youssef et al., 2021	NTS	50	8	16.00%	50	11	22.00%	1.38	[0.6; 3.13]	PCR
*bla OXA*	Elhariri et al., 2020	S. H	24	5	20.83%	9	1	11.11%	0.53	[0.07; 3.96]	PCR
											PCR
*bla OXA-61*	Divsalar et al., 2019	*C. jejuni*	80	42	52.50%	20	7	35.00%	0.67	[0.35; 1.25]	PCR
*bla SHV*	Youssef et al., 2021 and Elhariri et al., 2020	NTS and S. H	74	14	18.92%	59	12	20.34%	1.13	[0.49; 2.61]	PCR
*blaTEM*	Youssef et al., 2021, Besharati et al., 2020 and Elhariri et al., 2020	NTS, NTS, and S. H	149	19	12.75%	81	12	14.81%	0.91	[0.34; 2.44]	PCR
*Class 1 Integrons*	Besharati et al., 2020 and AbdEl-Aziz et al., 2020	NTS and *C. jejuni*	223	180	80.72%	42	29	69.05%	1.04	[1.01; 1.08]	PCR
*class 2 Integrons*	Besharati et al., 2020	NTS	75	22	29.33%	22	6	27.27%	0.93	[0.43; 2]	PCR
*cme B*	Divsalar et al., 2019	*C. jejuni*	80	41	51.25%	20	7	35.00%	0.68	[0.36; 1.29]	PCR
*tet(A)*	Divsalar et al., 2019	*C. jejuni*	80	17	21.25%	20	6	30.00%	1.41	[0.64; 3.11]	PCR
*tet(O)*	Divsalar et al., 2019 and Ghoneim et al., 2020	*C. jejuni*	84	71	84.52%	22	17	77.27%	0.92	[0.73; 1.16]	PCR

HN: Number of human isolates: HI: human isolates that have this gene; prevalance_H: prevalence among human isolates; AN: number of animal isolates; AI: animal isolates that have this gene; prevalance_A: prevalence among animal isolates; RR: relative risk; NTS: non-typhoidal *Salmonella*; S.H: *Salmonella* Heidelberg; *C. jejuni*: *Campylobacter jejuni.*

## Data Availability

The datasets generated during the current meta-analysis are available from the corresponding author upon reasonable request. All data analyzed for meta-analysis are included in the corresponding published articles, as reported in Appendix A.

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
