# Peer review of "Systematic Review and Meta-Analysis of Integrated Studies on Salmonella and Campylobacter Prevalence, Serovar, and Phenotyping and Genetic of Antimicrobial Resistance in the Middle East—A One Health Perspective"

_antibiotics, 2022, doi:10.3390/antibiotics11050536_

Round 1

Reviewer 1 Report

The paper is a good contribution both to better define the phenomena of antibiotic resistance for Campylobacter and Salmonella, and as a reference point for orienting the work related to this criticality.

Author Response

Comment: The paper is a good contribution both to better define the phenomena of antibiotic resistance for Campylobacter and Salmonella and as a reference point for orienting the work related to this criticality.

Response: Thank you for your feedback and we appreciate your comment.

Reviewer 2 Report

The authors' aim was to identify existing integrated One health studies on foodborne illnesses in the Middle East and to determine the prevalence, serovars, and antimicrobial resistence phenotypes and genotypes of Salmonella and Campylobacter strains; among humans and food producing animals;

The manuscript is entitled: Systematic review and meta-analysis of integreted studies on Salmonella and Campylobacter prevalence, serovar, and phenotyping and genetic of antimicrobial resistence in the Middle East - a One Health perspective

but, 26/41 studies (63,41%) were conducted in Egypt, and about 60% of Middle east countries have no published papers to be included in this review; for these reasons authors should change the title; for example: Systematic review and meta-analysis of integreted studies on Salmonella and Campylobacter prevalence, serovar, and pheno-typing and genetic of antimicrobial resistance in Egypt and other Middle Eastern countries - a One health perspective.

Throughout the text, tables and figures: Salmonella and Campylobacter in italics;

Throughout the text, tables end figures: Salmonella spp., Campylobacter spp.; (spp. not in italics)

Throughout the text: Salmonella Enteritidis, Salmonella Typhimurium, Salmonella Heidelberg; (Enteritidis, Typhimurium, Heidelberg: not in italics)

Line 135: Campylobacter

Line 214-215: (2% in diarrhea patients and 2% in high-risk population)....(2% diarrhea patients and 2% in high-risk population) - it's a repetition;

Line 200-201: 3.3. Prevalence and serotype distribution of Salmonella spp and campylobacter spp.......

Salmonella spp and Campylobacter spp.

I think the authors can improve the Discussion and Conclusions of their paper with more references regarding  antimicrobial resistance in Salmonella and Campylobacter; I would like to suggest a very recent EFSA publication, this is the link : https://efsa.onlinelibrary.wiley.com/doi/epdf/10.2903/j.efsa.2022.7209  

Author Response

Comment 1: The authors' aim was to identify existing integrated One health studies on foodborne illnesses in the Middle East and to determine the prevalence, serovars, and antimicrobial resistance phenotypes and genotypes of Salmonella and Campylobacter strains; among humans and food-producing animals;

The manuscript is entitled: Systematic review and meta-analysis of integrated studies on Salmonella and Campylobacter prevalence, serovar, and phenotyping and genetic of antimicrobial resistance in the Middle East - a One Health perspective.

but, 26/41 studies (63,41%) were conducted in Egypt, and about 60% of Middle East countries have no published papers to be included in this review; for these reasons, authors should change the title; for example Systematic review and meta-analysis of integrated studies on Salmonella and Campylobacter prevalence, serovar, and pheno-typing and genetic of antimicrobial resistance in Egypt and other Middle Eastern countries - a One health perspective.

Response: It is a very important point, and thank you very much for this comment; as mentioned above, the main objective is to systematically identify existing integrated One health studies on foodborne illnesses in the Middle East. Hence, when we built our inclusion criteria, we used the definition of the Middle East area based on UN agencies’ definitions and some scientific publications (1). We believe that the title should reflect the geography covered in the review process - not the findings. However, the reviewer raised an important point and we mentioned this important point already in the abstract, line 39.

Comment 2: Throughout the text, tables and figures: Salmonella and Campylobacter in italics;

Throughout the text, the tables end figures: Salmonella spp., Campylobacter spp.; (spp. not in italics)

Throughout the text: Salmonella Enteritidis, Salmonella Typhimurium, Salmonella Heidelberg; (Enteritidis, Typhimurium, Heidelberg: not in italics)

Line 135: Campylobacter

Line 214-215: (2% in diarrhea patients and 2% in high-risk population)....(2% diarrhea patients and 2% in high-risk population) - it's a repetition;

Line 200-201: 3.3. Prevalence and serotype distribution of Salmonella spp and campylobacter spp.......

Salmonella spp and Campylobacter spp.

Response: Thank you for your specific comments. We have taken all the comments into consideration and modified the text accordingly.

Comment 3: I think the authors can improve the Discussion and Conclusions of their paper with more references regarding  antimicrobial resistance in Salmonella and Campylobacter; I would like to suggest a very recent EFSA publication, this is the link: https://efsa.onlinelibrary.wiley.com/doi/epdf/10.2903/j.efsa.2022.7209  

 Response: Thank you for your comment and this recommendation (EU report). We have taken this reference and used it with other references in the introduction and discussion parts.

Reviewer 3 Report

This review evaluates the prevalence, serovars, and antibiotic resistance phenotypes and genotypes of Salmonella and Campylobacter among humans and food-producing animals in the Middle East using a meta-analysis approach from a One Health perspective. Overall, the study design is comprehensive and the manuscript provides valuable information.

However, the use of some terms (e.g. antimicrobial and antibiotic, Salmonella spp. and NTS, NST serovars) needs to be further clarified. There are too many formatting errors in the text. In addition, the authors need to focus more on interpretation of the results, not just cite what have been reported in the literature.

Some major points have been given below:

  1. Conceptually, Salmonella, non-typhoid Salmonella NTS, and NTS serovars (e.g., Salmonella Enteritidis, Salmonella Typhmurium) are from large to small. It’s better to introduce the background in this order.
  2. From the full text, the authors seem to focus more on NTS. Is the literature used by Salmonella and NST the same? because the number of NTS studies is not clear in the results section (Figure 2b and supplementary Tables). Please add this information.
  3. Lines 58-61: The data is from a decade ago, please add new data.
  4. Lines 251-255: Most phenotypic resistance had not a pooled RR close to 1, Please compare all with the Table 2a.
  5. The title format of 3.2. needs to be adjusted in the results section.
  6. Please increase the resolution of Figure1 and Figure 2. The legend is usually below the Figure. Please add the legend in Figure 2.
  7. It’s inappropriate to include references in the conclusion.
  8. Please use italics in the text: e.g. Salmonella, Campylobacter, gene names, I2    Please replace “Salmonella spp” with “Salmonella spp.”, replace “S. Enteritidis” with “S. Enteritidis” , the same is for other serovars.

Author Response

  1. Comment 1: Conceptually, Salmonella, non-typhoid Salmonella NTS, and NTS serovars (e.g., Salmonella Enteritidis, Salmonella Typhmurium) are from large to small. It’s better to introduce the background in this order.

Response: Thank you for your specific comments. We have considered this comment and modified the text accordingly.

  1. 2. Comment 2: From the full text, the authors seem to focus more on NTS. Is the literature used by Salmonella and NST the same? because the number of NTS studies is not clear in the results section (Figure 2b and Supplementary Tables). Please add this information.

Response: Yes, the reviewer is right, and it is essential to point out that we focused on the NTS. In the methodology part, we included only studies targeting NTS and excluded studies targeting S. Typhi and paratyphi (lines 121-122). We aimed to focus mostly on foodborne diseases from animal sources. Therefore, in figure 2b, these are the number of NTS studies. In addition, we have considered this comment and modified the text in line no. 190.

  1. 3. Comment 3: Lines 58-61: The data is from a decade ago, please add new data.

   Response: Thank you, we have considered this comment and added new data in the introduction.

  1. Comment4: Lines 251-255: Most phenotypic resistance had not a pooled RR close to 1, Please compare all with Table 2a.

Response: Thank you very much this point was missed and, we have considered this comment and modified the text to be “While, the pooled RR close to 1 in ampicillin and streptomycin suggesting a similar probability of occurrence in humans and animals. For the other antibiotics, no clear pattern was detected”.

  1. Comment5: The title format of 3.2. needs to be adjusted in the results section.

Response: Thank you, we have considered this comment and the title has been adjusted.

  1. Comment6: Please increase the resolution of Figure1 and Figure 2. The legend is usually below the Figure. Please add the legend in Figure 2

Response: Thank you, we have increased the resolution of the Figures and put the legend below the Figures. We added a legend to Figure 2.

  1. Comment7: It’s inappropriate to include references in the conclusion.

Response: Thank you, we have considered this comment and removed the references in the conclusion.

  1. Comment8: Please use italics in the text: e.g. SalmonellaCampylobacter, gene names, I2    Please replace “Salmonella spp” with “Salmonella spp.”, replace “S. Enteritidis” with “S. Enteritidis” , the same is for other serovars.

Response: Thank you, we have considered this comment and modified the text accordingly.
